# Mining Exploratory Queries for Conversational Search

## ABSTRACT

Users' queries are usually vague, and their search intents tend to be ambiguous, thereby needing clarification. Search clarification has been proposed as an important technique to clarify users' current search intent by asking a clarifying question and providing several clickable sub-intent items as clarification options. However, in addition to drilling down the current query, users may also have *exploratory needs* that diverge from their current search intent. For example, a user searching for the query "Cartier women watches" may also potentially want to explore some parallel information by issuing queries such as "*Rolex* women watches" or "Cartier women *bracelets*", named *exploratory queries* in this paper. These exploratory needs are common during the search process yet cannot be satisfied by current search clarification approaches which typically stick to the sub-intents of the current query. This paper focuses on mining exploratory queries as additional clickable options to meet users' exploratory needs in conversational search systems. Specifically, we first design a rule-based model that generates exploratory queries based on the current query's top retrieved documents. Then, we propose using the data generated by the rule-based model to train a neural generation model through multi-task learning for further generalization. Finally, we borrow the in-context learning ability of the large language model to generate exploratory queries based on prompt engineering. We conduct an extensive set of experiments and the results show that our proposed methods generate higher-quality exploratory queries compared with several baselines. The results also demonstrate that the structure information in top retrieved documents is useful for generating exploratory queries.

## 1 INTRODUCTION

Conversational search [33, 38] is a natural language-based search approach to help users obtain information from the Web, using a conversational interface to realize human-like communication. In a conversational search system, users' queries tend to be ambiguous or faceted [3], hindering the search engine from understanding the user's potential search intent. Search clarification [35, 43, 47] has been proposed as an effective way to mitigate this issue. The grey part in Figure 1 shows an example of search clarification. After a user issues the query "Cartier women watches", the search engine will ask a *clarifying question* and provide several *clarification items* of sub-intents such as "Cartier women watches price" to clarify the user's search intent. Search clarification focuses on specializing the

*Conference'17, July 2017, Washington, DC, USA*
© 2023 Association for Computing Machinery.
ACM ISBN 978-x-xxxx-xxxx-x/YY/MM...$15.00
https://doi.org/10.1145/nnnnnnn.nnnnnnn

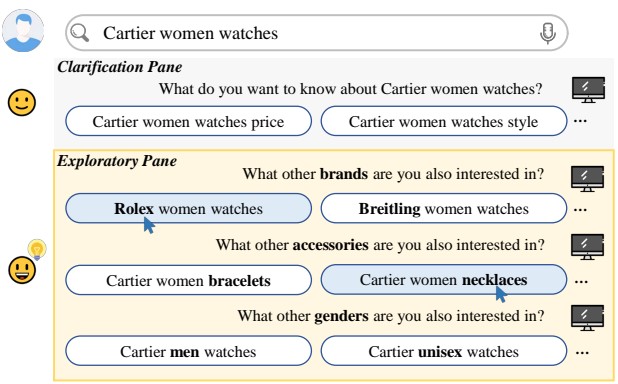

**Figure 1: An application of our exploratory queries in search clarification. The grey part is the search clarification pane and the yellow part represents our exploratory search pane.**

query by appending a term, like "price" and "style" in Figure 1 for clarifying users' potential search intents. Existing studies usually mine these clarification items from the query log [43] by applying some query suggestion techniques [1, 5, 7, 24].

However, in addition to such *clarification need*, users may also have *exploratory needs* [4, 11, 19, 29] in some cases. For example, when an applicant aiming at studying abroad issues "study abroad **resume**", she may also be interested in other relevant materials by issuing further queries "study abroad **recommendation letter**" or "study abroad **personal statement**". For another example, a user who searches for "**Cartier** women watches" is likely to issue exploratory queries such as "**Rolex** women watches" or "**Breitling** women watches", to compare different brands before making a purchase decision. These exploratory queries are different from the clarification items in existing search clarification studies. They modify a term[1] included in the original query to horizontally represent its parallel intents.

We believe that mining these exploratory queries in the context of conversational search is important. **First**, as mentioned in existing studies [4, 11, 19, 29], users' exploratory search behaviors are common in real-world search systems. Boldi et al. [4] reported that users' exploratory search behaviors constituted 48-56% of a Yahoo! search log, even bigger than clarification behaviors (30-38%). Therefore, displaying an additional exploratory pane (yellow part in Figure 1) is a good extension of the existing search clarification in conversational search scenario. **Second**, these exploratory queries provide new topics or broader information space to users, which enhances the diversity of clickable options and improves the users' search experience from new perspectives.

Despite the importance and usefulness of exploratory queries, it is less emphasized in previous studies of conversational search. In this paper, we make the first step to mining exploratory queries as the clickable options to meet users' exploratory needs, extending the scenario of mining sub-intents as the clickable options in the

---

[1]In this paper, we define that a term is a word or a phrase in a query.

clarification pane shown in Figure 1. To this end, we first define the exploratory query as the reformulation of the original query with *replacing a term in the original query with another term*. For example, the query "Cartier women watches" may be reformulated as "Rolex women watches". This is different from search clarification which just focuses on specializing the query by appending a term after the query for mining the query's sub-intents (like "Cartier women watches *price*"). On the other hand, since a query contains multiple keywords, it may correspond to multi-group exploratory queries. Taking the query "Cartier women watches" as an example, replacing the terms "Cartier", "women" and "watches" respectively results in other brands of women watches (e.g., "Rolex women watches", "Breitling women watches"), Cartier watches for different genders (e.g., "Cartier men's watches", "Cartier unisex watches") and other accessories for women (e.g., "Cartier women bracelets", "Cartier women necklaces"). We believe that presenting exploratory queries in groups delivers more comprehensive and understandable exploratory intents for users, compared with clarification items presented in a flat list. Besides, it also allows the system to ask an *exploratory question* for each group as shown in Figure 1 to improve users' search experience and invoke their exploratory behaviors. In this paper, we focus on generating the exploratory queries and leave the generation of exploratory questions as future work.

The key to generating exploratory queries is to mine multiple groups of terms parallel to the ones occurring in the original query. We observe that these parallel terms are usually organized in *list styles* in the query's top retrieved documents. For example, in the top retrieved documents of the query "Cartier women watches", "watches" will be listed with other types of accessories (e.g., "bracelets") using "<li>" tags in the Cartier official website. Besides, to help users filter watches by gender, "women" will be listed with "men" and "unisex" under the "<select>" tag. Due to the design of HTML pages, list structures contained in top retrieved documents naturally contain parallel information, which has been revealed by previous studies [9, 10, 20, 47]. **In this paper, we propose leveraging the list structures in top retrieved documents to generate multi-group exploratory queries**. Due to the lack of large-scale human-annotated exploratory query data required for training a parallel term generation model, we first design a Rule-based Parallel Reformulation model **RPR** that extracts list items from top retrieved documents, uses them to conduct parallel reformulation to obtain exploratory query candidates, and then ranks these candidates based on various human-designed features. Besides, since rule-based methods are prone to suffer from low generalization ability, we further train an Exploratory Query Generation model **EQG** using the data generated by RPR as weak supervision signals for further generalization. We design the generation task and an additional classification task to ensure the quality of generated results in a multi-task learning manner. Finally, by borrowing the strong few-shot and in-context learning ability of the large language models (LLMs), we propose another LLM-based exploratory query Generation method **LLMG** based on prompt engineering. We reveal that even with an LLM, the list items extracted by RPR are also essential for exploratory query generation.

To evaluate our models, we aggregate the human-written and model-generated exploratory queries into a pool and employ three annotators to judge the quality of these exploratory queries in a pool-based manner to construct the evaluation data[2]. The results show that the rule-based parallel reformulation model RPR significantly outperforms several baselines and the exploratory query generation model EQG further improves the results of RPR. Additionally, our proposed LLMG outperforms all the models with the support of list items extracted from top retrieved documents. We further perform an ablation study to prove the effectiveness of each component in our models and conduct a case study to intuitively compare the exploratory queries generated by different methods.

The main contributions are summarized as follows:

(1) We propose to mine exploratory queries to meet users' exploratory needs in conversational search.

(2) We design a rule-based parallel reformulation model to generate exploratory queries based on list items extracted from top retrieved documents.

(3) Based on extracted list items, we further propose a generative model trained with multi-task learning and an LLM-based generation method with prompt engineering.

(4) We build an evaluation dataset, design evaluation approaches, and conduct extensive experiments to demonstrate the effectiveness of our proposed methods.

## 2 RELATED WORK

### 2.1 Search Intent Mining

Search clarification, query suggestion and query facets mining are popular approaches to assist users in expressing their potential information needs. Search clarification [35, 43, 47] provides a pane consisting of several clickable options and a clarifying question to clarify users' potential information needs of the ambiguous or faceted query. Query suggestion [1, 5, 7, 8, 16, 24, 31, 36], as a core utility for many industrial search engines, aims to recommend a set of relevant or alternative queries that are likely to be clicked by users. Query auto completion (QAC) [6, 18, 32, 39] displays a drop-down list of suggested queries based on the partial query text entered by the user. It focuses on additive reformulations on the user query, which can also be considered a kind of query suggestion. Query facets mining [9, 10, 17, 20, 21] is a technique that identifies and extracts different facets of users' search queries to help specify their search intents. The techniques mentioned above mainly focus on specializing users' queries by appending some terms after the query to *clarify their sub-intents*. However, in addition to sub-intents, users may also desire to explore other contents beyond their current information needs, which we refer to as exploratory needs in Section 1. In this paper, we deviate from these traditional methods that stick to users' sub-intents and focus on recommending exploratory queries to satisfy users' exploratory needs.

### 2.2 Exploratory Search

Exploratory search [41] is a complex information-seeking process to tackle the situation when the user's information need is vague or not well-defined. It has been studied by many works. Awadallah et al. [15] proposed to build an association graph to help users explore and complete some complex search tasks. Ksikes et al. [22] designed a faceted search system for exploratory search. Lissandrini

---

[2]The evaluation data is available at: https://anonymous.4open.science/r/EvalData

et al. [25] produced query suggestions based on knowledge graph to help users do exploratory searches. Previous studies [11, 19] have also shown that users usually have exploratory interests and tend to reformulate their search queries in a parallel way. Besides, Ma et al. [29] observed and analyzed users' search logs and found that there are many search goal shift phenomena in the exploratory search process. Inspired by these works, in this paper, we focus on generating exploratory queries to further improve users' exploratory searches in conversational search systems.

## 2.3 Structured Information in Search Results

Search results often contain rich and contextual structured information which have been utilized in many relevant studies. For example, Dou et al. [9, 10] and Kong et al. [20, 21] mined query facets by extracting list structures from search results. Guo et al. [12] proposed utilizing hierarchical structures in HTML to pre-train a language representation model, which can then be fine-tuned for ad-hoc document retrieval. Additionally, Zhao et al. [46, 47] demonstrated the effectiveness of using list structures extracted from search results to generate high-quality clarifying questions. Overall, these studies demonstrate the importance of utilizing structured information in HTML documents, especially list-shaped structures, for various tasks. We deem that these list structures in top retrieved documents usually illustrate parallel information and are also helpful for generating exploratory queries in our studies.

## 3 METHODS

As we mentioned in Section 1, top retrieved documents contain rich list structures that help to generate exploratory queries. In this section, we propose three methods (RPR, EQG, and LLMG, shown in Figure 2) for exploratory query generation. In RPR, we first retrieve top-$n$ documents of the original user query $q_o$, then extract all list structures and plain texts from these documents. After that, we perform parallel reformulation for $q_o$ using the extracted list items to obtain a set of exploratory query candidates and rank all these candidates based on various human-designed features. To improve the generalization of the generated exploratory queries, we further propose EQG which is a BART-based weakly supervised generation model trained with multi-task learning strategy. We also propose an LLM-based method LLMG to generate exploratory queries based on our well-designed prompts.

## 3.1 Rule-based Parallel Reformulation

The RPR algorithm uses the list items extracted from top retrieved documents to reformulate the original user query $q_o$, so as to obtain corresponding exploratory query candidates. It then uses various human-designed features to rank the candidates and finally divide them into different groups. The algorithm consists of four components: (1) Top Documents Retrieval (2) Lists and Texts Extractor (3) Parallel Reformulation and Ranking (4) Candidates Grouping.

*3.1.1 Top Documents Retrieval.* We first obtain the top-$n$ search results (snippets, document URLs, etc.) of the original user query $q_o$ using Bing's Web Search API. Then, we crawl the HTML file corresponding to each document URL for further lists and plain text extraction.

*3.1.2 Lists and Texts Extractor.* As mentioned in Section 1, the list structures in HTML documents usually illustrate parallel relations. For example, "watches", "bracelets" and "necklaces" (all belonging to the accessories of Cartier) will be listed together using "<li>" tags in an HTML page. Intuitively, when a query term appears in one list, the other items in this list are likely to be parallel to this term. Thus, these list items are important data resources for parallel reformulation. Additionally, the plain texts in HTML documents are paragraphs that contain contextual unstructured natural language information, which is highly relevant to the user query and may also contain exploratory queries (see *Texts* extracted by RPR in Figure 2). Thus, in addition to list structures, we also extract the plain texts in documents as auxiliary information for parallel reformulation. We implement the efficient and effective algorithm proposed in [9] to extract list structures from HTML tags, repeat regions, etc. We represent all the extracted lists as $L = \{L_1, L_2, \ldots, L_M\}$, where each list contains several items $L_i = \{L_{i,1}, L_{i,2}, \ldots, L_{i,m}\}$. We simultaneously extract HTML plain texts $T = \{T_1, \ldots, T_i, \ldots, T_K\}$ for each retrieved document, where $T_i$ denotes the concatenated text of all paragraphs in the $i$-th document.

*3.1.3 Parallel Reformulation and Ranking.* In this part, we use the list items extracted above to generate exploratory query candidates and rank them based on various manual features. We first gather all the items from all extracted lists in $L$ to obtain the whole item set $I$. Then we replace each term in the original query with each item in $I$, obtaining the corresponding exploratory query. For example, an exploratory query "Cartier women bracelets" is obtained by replacing "watches" in the query "Cartier women watches" with the item "bracelets". The process of parallel reformulation can be reformulated as: $q_o \xrightarrow{e,t} q$, which denotes that exploratory query $q$ is obtained by replacing the term $t$ in original query $q_o$ with item $e$. We apply CoreNLP [30] tool to extract the terms in $q_o$ and filter out those terms with no practical meaning, such as conjunctions, prepositions, and function words. We design various features for each exploratory query candidate $q$ including list co-occurrence feature $F^l$, concept feature $F^c$, popularity feature $F^p$, and item feature $F^i$. In the rest of this section, we will introduce their definitions and calculations.

(1) **List Co-occurrence Feature** $F^l$. Items in the same list often share similar characteristics and are conceptually parallel to each other (such as "watches", "bracelets", and "rings"). Intuitively, the more frequently item $e$ and replaced term $t$ appear in the same list, the stronger the conceptual parallel between them and the greater the likelihood that the exploratory query $q$ is a suitable candidate for $q_o$. Thus, we have:

$$F^l(q) = a_l \cdot \tanh\left(b_l \cdot \sum_i \text{occ}(e, t, L_i)\right), \quad (1)$$

where $a$ and $b$ are the adjustment coefficient and importance coefficient respectively, and $\tanh(\cdot)$ is used to control the weight of each feature. $\text{occ}(A, B, C)$ is a binary function, where $\text{occ}(A, B, C) = 1$ when $A$ and $B$ both occur in $C$. Otherwise, $\text{occ}(A, B, C) = 0$.

(2) **Concept Feature** $F^c$. In addition to list structures, knowledge graphs like Concept Graph [40, 42] can also help judge whether the item $e$ is conceptually similar to the replaced term $t$ (note that $q$ is obtained by replacing term $t$ in $q_o$ with item $e$). In Concept

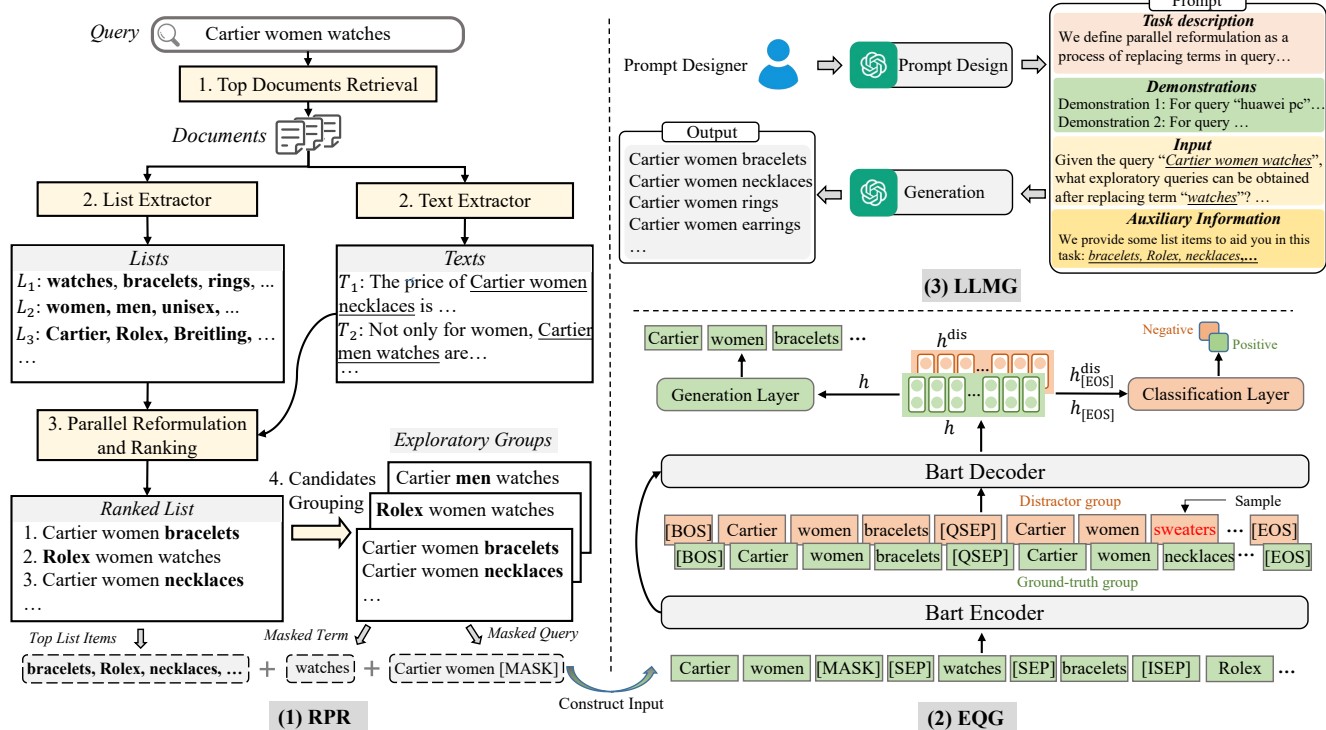

**Figure 2: An overview of our proposed models (1) RPR, (2) EQG, and (3) LLMG. RPR generates exploratory queries based on list structures and plain texts extracted from the top retrieved documents. Then, we fine-tune EQG using data generated by RPR through a mask-and-fill strategy. An classification task is designed to enhance the generation task in the training stage. Finally, we introduce the LLMG method, which uses well-designed prompts to generate exploratory queries based on LLM.**

Graph, each instance usually corresponds to multiple concepts. For example, the concepts of "Cartier" include company, brand, jewelry brand, watch brand, etc. Therefore, the more similar the concept set of item $e$ and replaced term $t$ are, the more likely they are parallel to each other. Then the Concept Feature $F^c$ can be calculated as:

$$F^c(q) = a_c \cdot \tanh \left( b_c \cdot \frac{C(e) \cap C(t)}{C(e) \cup C(t)} \right), \quad (2)$$

where $C(A)$ denotes the concept set of $A$ in Concept Graph.

(3) **Popularity Feature** $F^p$. A more popular exploratory query is more likely to be clicked by users. Thus, we also compute the popularity feature for each candidate $q$. We claim that contents of retrieved documents naturally contain popular information and candidates that frequently occur in the plain texts of top retrieved documents are more likely to be popular with users. Then feature $F^p$ can be formulated as follows:

$$F^p(q) = a_p \cdot \tanh \left( b_p \cdot \sum_i N(q, T_i) \right), \quad (3)$$

where $N(A, B)$ denote the frequency of $A$ occurring in $B$. Since the length of extracted plain texts can be very large, we implement the function $N(A, B)$ based on an efficient string-searching algorithm Aho–Corasick [2].

(4) **Item Feature** $F^i$. If a candidate $q$ appears as a list item in $L$, then it is more likely for $q$ to be a useful and faithful exploratory

query to the user. However, $q$ (*e.g.*, "Cartier unisex watches") sometimes does not match any list item exactly, but its terms can appear in some list items (*e.g.*, "unisex Cartier watches", "Cartier watches for unisex"), which can also prove the usefulness and faithfulness of $q$. Thus an exploratory query $q$ with more terms appearing in any list item is more likely to be a useful and faithful candidate. Therefore the item feature $F^i$ is defined as follows:

$$F^i(q) = a_i \cdot \tanh \left( b_i \cdot \max_{j,k} \frac{q \cap L_{j,k}}{|q|} \right), \quad (4)$$

where $|q|$ means the word set size of $q$ and $L_{j,k}$ means the $k$-th item in the $j$-th list.

We add these features together to get the final score of each candidate $q$ and rank all the candidates based on their final scores:

$$\text{score}(q) = F^l(q) + F^c(q) + F^p(q) + F^i(q). \quad (5)$$

To ensure the quality of generated exploratory queries, we set a threshold $\tau$ to filter out candidates with $\text{score}(q) \leq \tau$, where $\tau$ is a hyperparameter.

*3.1.4 Candidates Grouping.* We divide the exploratory queries in the ranked list into different groups according to their replaced term $t$. For each group $Q$, following previous studies of search clarification [43, 44], we only keep the top-5 candidates as the final exploratory queries. Finally, we obtain $Q = \{q_1, \ldots, q_k\}$, where $k \leq 5$ and $q_i$ represents the $i$-th exploratory query. The grouping

method based solely on replaced terms may have limitations due to its simplicity. In the future, we will explore more advanced grouping methods, which will be discussed in Section 5.6.

## 3.2 Exploratory Query Generation Model

The generation of RPR merely relies on the search results and several human-designed features, which could suffer from data sparsity problems in some cases. For example, when the query's search results contain few relevant documents, RPR may fail to extract useful list items for exploratory query generation. To mitigate this issue, we intend to use the pre-trained language model BART [23] to improve the generalization ability of RPR. The reason is that: our exploratory query generation task can be treated as a mask-and-fill problem (replacing the term in the query with a "[MASK]" token and filling the "[MASK]" with a term which is parallel to the replaced one). While the BART model (implicitly captured a large amount of knowledge) could fill the "[MASK]" with more appropriate terms after fine-tuning. Thus, we intend to design a BART-based language model EQG that generates exploratory queries based on the masked query and further generalizes RPR.

Intuitively, the reformulated parts of the exploratory queries in one group should have *conceptual consistency*. For example, in the exploratory group [Cartier women bracelets, Cartier women necklaces, Cartier women rings], the terms "bracelets", "necklaces", and "rings" all belong to the accessories of Cartier. To further improve the quality of generated results from a *group level*, we propose to use an additional classification task that distinguishes the ground-truth group from a distractor group. The framework of EQG is illustrated in Figure 2 (bottom right).

*3.2.1 Generation Task.* In this part, EQG aims to generate a group of exploratory queries through a mask-and-fill strategy. For each group $Q$ extracted by RPR, we replace the term to be reformulated in the original query $q_o$ with a "[MASK]" token to obtain corresponding masked query $q_o^m$ (see Figure 2). Then we provide two additional pieces of information to prompt model the potential terms that can be used to fill in the "[MASK]" token: (1) The masked term (denoted as $t^m$); (2) The list items contained in top-$v$ exploratory queries in the ranked list (such as bracelets, rolex, etc.), denoted as $S = \{s_1, \ldots, s_v\}$ ($v$ is set as 100 in this paper). Then the BART learns to fill the masked query $q_o^m$ with terms that are conceptually parallel to the masked term $t^m$. We concatenate the masked query $q_o^m$, masked term $t^m$ and top-$v$ list items $S$ together as the input to the BART encoder (separated by "[SEP]"):

$$E^i = q_o^m \text{ [SEP] } t^m \text{ [SEP] } s_1 \text{ [ISEP] } s_2 \text{ [ISEP] } \ldots s_v, \quad (6)$$

where "[ISEP]" is used to separate list items. We concatenate all exploratory queries in $Q$ using a special token "[QSEP]" as the generation target. Then we borrow the cross-entropy loss from Seq2Seq model [37] to calculate the generation loss $\mathcal{L}_{gen}$ as:

$$\mathcal{L}_{gen} = \min \sum_{i=1}^{|Q|} \sum_{j=1}^{|q_i|} -\log p(q_{i,j}|E^o, q_{i,1}, \ldots, q_{i,j-1}), \quad (7)$$

where $E^o$ denotes the BART encoder's output.

*3.2.2 Classification Task.* As discussed, the exploratory group should have conceptual consistency. In this part, we design a classification

task on the BART decoder by allowing the model to distinguish between the ground-truth group and the distractor group, thereby improving the quality of generated exploratory queries.

To obtain the distractor group for each group $Q$, we randomly select one exploratory query in $Q$ and replace the term to be reformulated with a randomly sampled list item (e.g., "sweaters") from the whole item set $I$ (see Figure 2). The BART decoder inputs two sequences (*i.e.*, the concatenation of exploratory queries in ground-truth group and distractor group) and outputs their hidden states $h$ and $h^{dis}$ respectively. Then we pass the representations of their "[EOS]" token through a classification layer to get two values $v$ and $v^{dis}$:

$$v = \text{MLP}(h_{[EOS]}), \qquad v^{dis} = \text{MLP}(h_{[EOS]}^{dis}). \quad (8)$$

Finally, we provide a binary label and apply a cross-entropy loss to calculate the classification loss:

$$\mathcal{L}_{cls} = -\log \frac{\exp(v)}{\exp(v) + \exp(v^{dis})}. \quad (9)$$

The final training objective combines the generation loss $\mathcal{L}_{gen}$ and classification loss $\mathcal{L}_{cls}$:

$$\mathcal{L} = \mathcal{L}_{gen} + \lambda \mathcal{L}_{cls}, \quad (10)$$

where $\lambda$ is a weight parameter to control the ratio of learning for the classification task.

*3.2.3 Inference.* For each query, we first run RPR to obtain the masked query $q_o^m$ and masked term $t^m$ for each exploratory group and the top-$v$ list items $S$. Then, EQG generates each group based on corresponding masked query $q_o^m$, masked term $t^m$, and top-$v$ list items $S$. We use beam search to collect exploratory queries for each group and keep a maximum of 5 exploratory queries in each group.

## 3.3 LLM-based Exploratory Query Generation

In this part, we design LLMG to further validate the effects of extracted list items on exploratory query generation. We designed task-specific prompts to instruct an LLM to generate one group of exploratory queries at a time (same as EQG in Section 3.2).

*3.3.1 Prompt Design.* Previous study has shown that the performance of LLMs tends to be sensitive to the design of prompts [28]. To make our prompt more robust, we design our prompts in two steps following [26]: (1) Prompt Description. We design the original prompt which includes our task description, several demonstrations, and the input which consists of the original query (e.g., "Cartier women watches") and the term to be replaced (e.g., "watches") provided by RPR. Besides, we add the top list items of the query (extracted by RPR) to the prompt as auxiliary information. (2) Multi-Step Optimizations. We provide the LLM with our prompt respectively and ask "Could you give me some advice on improving the prompt?", and optimize the prompts according to the LLM's suggestions. Besides, we also test the prompt on some samples, analyze the quality of the generated results, and then further optimize the prompt. We perform these two optimization strategies iteratively until the quality of the generated results no longer improves.

*3.3.2 Generation.* Similar to EQG, we first run RPR to obtain the top list items and the exploratory groups which indicate the terms that could be replaced in the original query. Then we construct the prompt and feed it to the LLM for generation. We tell LLM to generate at most 5 queries at a time while ensuring the usefulness, faithfulness, and readability of the results. To further prove the effectiveness of the list items in generation, we abandon the top list items (*i.e.*, Auxiliary Information) from the current prompt to obtain a new prompt for LLM generation, denoted as LLMG (-list).

## 4 EXPERIMENTAL SETTINGS

### 4.1 Data

MIMICS [44] is a search clarification dataset that includes a large number of real Web search queries sampled from Bing query logs. In this paper, we intend to sample queries from MIMICS for the training, validation, and evaluation of our models. For the training data of EQG, we randomly sample 40k queries from MIMICS and generate about 60k pieces of weakly supervised training data using RPR, approximately 1.5 terms replaced for each query on average.

As we do not find any publicly available dataset for the validation and evaluation of our task, we randomly sample 50 and 100 queries from MIMICS dataset to build our validation and evaluation data in a pool-based manner. We ensure that there is no overlap between the queries used for training, validation, and evaluation. For each query, we first ask a subject to manually write corresponding exploratory queries after a comprehensive survey on some resources (such as Wikipedia and top retrieved documents from Bing). Then we aggregate exploratory queries generated by all models (including the ablation models in Section 5.2) we want to evaluate and the human-written ones to form a pool. Note that such aggregation follows the classic Cranfield experiments [27], which aims to ensure a fair evaluation of all models.

We employ three annotators who understand our task well to help select high-quality exploratory queries to construct the final ground truth. Specifically, we ask the annotators to evaluate these exploratory queries in terms of three aspects: **usefulness**, **faithfulness**, and **readability**. We give them the definition and some examples in Table 1 to help them better understand these aspects. If an exploratory query satisfies all three aspects, then a *Good* label should be given. Otherwise, they are asked to give a *Bad* label. We also provide the annotators with the Bing search results of each query to help them better judge the faithfulness of each exploratory query. The final label of each exploratory query is determined by the majority vote among the three annotators. The value of Fleiss' kappa among the three annotators is 91.63%, which shows an almost perfect agreement. Finally, we manually create exploratory groups for each query (such as "Cartier women [MASK]" for the query "Cartier women watches") and assign the exploratory queries whose final labels are **Good** in the pool to corresponding created groups as the ground-truth.

### 4.2 Evaluation Metrics

To evaluate the generated multi-group exploratory queries, we first assign each output group $Q$ to a ground-truth group $Q'$ which covers the maximum number of exploratory queries in $Q$. Then, for each group pair $(Q', Q)$, we adopt four sets of evaluation metrics

which have been widely adopted in aspect items generation task [13, 14, 34]: (1) **Term overlap**. These metrics include Term Precision, Term Recall, and Term F1-measure that have been applied for the evaluation of query facet extraction models [20]. (2) **Exact match**. They calculate the precision, recall, and F1-measure of generating the exact exploratory query that appears in the ground-truth group. (3) **Set BLEU score**. BLEU is defined to evaluate the similarity between a single candidate text and a group of references. We implement the Set BLEU score for 1-gram and 2-gram to measure the lexical similarity between the generated exploratory group (*i.e.*, $Q$) and the ground-truth group (*i.e.*, $Q'$). (4) **Set BERT-Score**. Since the above three evaluation metrics just measure the lexical similarity, following previous studies, we also implement Set BERT-Score [45] to compute the semantic similarity between $Q'$ and $Q$. We calculate the mean performance of all generated groups for each mentioned metric, which serves as the final result for each query.

### 4.3 Implementation Details

We apply Bing Search API v7 to obtain the top-n search results for RPR in Section 3.1.1, and the number of search results $n$ is 50.[3] The parameters of the RPR model (including adjustment coefficient $a_x$, importance coefficient $b_x$ and the threshold $\tau$) are tuned by grid search with the step of 0.1 on the validation set. We apply Pytorch to implement the EQG model and initialize its parameters based on the pre-trained BART-base model. In all experiments, the batch size is set as 6, and the max length of the input and output is set as 512 and 64 respectively. The weight parameter $\lambda$ is set as 1.0. We use AdamW optimizer to optimize the model with a learning rate of $5 \times 10^{-5}$. We utilize GPT-3.5-turbo with OpenAI API for LLMG.[4]

## 5 EXPERIMENTAL RESULTS

### 5.1 Overall Results

In this part, we choose two BART-based models applied in previous studies [34] as our baselines for comparison: (1) BART (q). It only takes the concatenation of masked query $q_o^m$ and the replaced term $t^m$ as input and generates the exploratory queries. (2) BART (qs). BART (qs) uses the concatenation of masked query $q_o^m$, masked term $t^m$, and snippet texts as input for generation. Previous studies [13, 34] have shown that search result snippets contain rich semantic information which helps better understand the user query. Like EQG, the two baselines are also trained using the data generated by RPR. Our aim in comparing with BART (qs) is to demonstrate the effectiveness of extracted list items on exploratory query generation. The experimental results are shown in Table 2.

The results show that: (1) Our RPR outperforms the two BART-based baselines and our EQG further improves RPR on all evaluation metrics. This illustrates that, compared with the rule-based method RPR, our well-designed EQG has a better ability to find more appropriate parallel terms, and thus generate higher-quality exploratory queries. (2) Our BART-based model EQG outperforms BART (qs) significantly, illustrating that the list structures extracted from search results are more effective in generating exploratory queries compared with snippet texts. (3) Our LLM-based method LLMG consistently outperforms all the methods with p-value <

---

[3]Bing Search API: https://www.microsoft.com/en-us/bing/apis/bing-web-search-api
[4]GPT-3.5-turbo: https://platform.openai.com/playground?model=text-davinci-003

**Table 1: The definition of evaluation aspects and several examples of Good and Bad exploratory queries.**

| Aspect | Definition |
|---|---|
| Usefulness | The exploratory query is parallel to the original user query and can meet users' exploratory needs. |
| Faithfulness | The exploratory query should make sense and provides faithful information that users can trust. |
| Readability | The exploratory query is free of grammatical errors, smooth and easy to understand. |
| **Examples** | |

| Query: *shirts for men* | | Query: *Cartier women watches* | |
|---|---|---|---|
| Good | jackets for men, pants for men | Good | Cartier women necklaces, rolex women watches |
| Bad (useless) | **cars** for men, **white shirts** for men | Bad (useless) | Cartier **ladies** watches |
| Bad (unfaithful) | **best** for men, **kids** for men | Bad (unfaithful) | Cartier women **shirts**, Cartier women **sweaters** |
| Bad (unreadable) | jackets **for men for men**, **jack** for men | Bad (unreadable) | Cartier **women women** necklaces |

**Table 2: Exploratory query generation evaluation results and ablation studies. "†" indicates the model outperforms the best baseline significantly with paired t-test at $p$-value < 0.05 level. The best results are in bold.**

| Model | Term Overlap | | | Exact Match | | | Set BLEU | | Set BERT-Score | | |
|---|---|---|---|---|---|---|---|---|---|---|---|
| | Prec | Recall | F1 | Prec | Recall | F1 | 1-gram | 2-gram | Prec | Recall | F1 |
| BART (q) | 0.4916 | 0.2092 | 0.2816 | 0.3803 | 0.1244 | 0.1778 | 0.2362 | 0.1293 | 0.2348 | 0.2361 | 0.2354 |
| BART (qs) | 0.5404 | 0.2426 | 0.3235 | 0.4453 | 0.1564 | 0.2222 | 0.2574 | 0.1588 | 0.2468 | 0.2472 | 0.2470 |
| RPR | 0.6354 | 0.2836 | 0.3768 | 0.5749 | 0.2169 | 0.2981 | 0.2824 | 0.2000 | 0.2702 | 0.2698 | 0.2700 |
| EQG | 0.6628 | 0.3043 | 0.4037 | 0.5934 | 0.2241 | 0.3101 | 0.3179 | 0.2058 | 0.2847 | 0.2842 | 0.2844 |
| LLMG (-list) | 0.6521 | 0.3199 | 0.4139 | 0.6087 | 0.2540 | 0.3453 | 0.3316 | 0.2109 | 0.2703 | 0.2708 | 0.2706 |
| LLMG | **0.7057**† | **0.3352**† | **0.4382**† | **0.6690**† | **0.2703**† | **0.3747**† | **0.3587**† | **0.2293**† | **0.2936** | **0.2939** | **0.2937** |
| RPR w/o. $F^l$ | 0.5047 | 0.2249 | 0.2938 | 0.4128 | 0.1489 | 0.2014 | 0.2379 | 0.1494 | 0.2251 | 0.2248 | 0.2249 |
| RPR w/o. $F^c$ | 0.5488 | 0.2527 | 0.3300 | 0.4540 | 0.1723 | 0.2338 | 0.2459 | 0.1639 | 0.2469 | 0.2465 | 0.2467 |
| RPR w/o. $F^p$ | 0.6198 | 0.2467 | 0.3379 | 0.5652 | 0.1824 | 0.2604 | 0.2214 | 0.1758 | 0.2293 | 0.2289 | 0.2291 |
| RPR w/o. $F^i$ | 0.6012 | 0.2553 | 0.3404 | 0.5313 | 0.1877 | 0.2580 | 0.2366 | 0.1697 | 0.2385 | 0.2382 | 0.2383 |
| EQG w/o. CT | 0.6232 | 0.2735 | 0.3683 | 0.5455 | 0.1911 | 0.2709 | 0.3011 | 0.1775 | 0.2727 | 0.2725 | 0.2726 |

0.05 on most metrics, which indicates the significance of the improvements. In addition, after removing the list items from the prompt, there is a noticeable performance drop on LLMG (-list), which proves that the extracted list items can help LLM generate better exploratory queries.

## 5.2 Ablation Studies

One of our main conclusions is that the human-designed features used in RPR are important to generate high-quality exploratory queries as weak supervision signals to train EQG. To prove the effectiveness of these features, we conduct an ablation study by removing the four features mentioned in Section 3.1 one by one (denoted as RPR w/o $F^x$). We also drop the classification task from EQG (denoted as EQG w/o. CT) to demonstrate its effectiveness. The results are shown in the bottom part of Table 2.

It can be seen that removing any component will damage the results on all evaluation metrics. As for RPR, abandoning the List Co-occurrence Feature $F^l$ causes the most decline in almost all metrics, which further confirms that list structures in search results contain important information for parallel reformulation. Similar to $F^l$, removing Concept Feature $F^c$ also results in an obvious drop in the evaluation metrics. This is because the structure knowledge in Concept Graph also contributes to measuring the parallel relationship. Besides, the Popularity feature $F^p$ and Item feature $F^i$

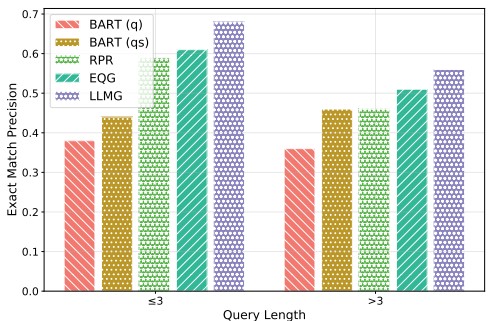

**Figure 3: Experiments on different query lengths.**

also play an important role in RPR by judging the quality of each candidate exploratory query. As for EQG, after abandoning the classification task, its performance also declines. For example, it declines by about 5.8% on term overlap precision and 8.04% on exact match precision. In summary, the ablation study proves the effectiveness of each component in our models.

## 5.3 Experiments with Different Query Lengths

In this section, we intend to investigate the performance of our models on queries of different lengths. Due to the average length of our evaluated queries being 2.64, we divide the evaluated queries

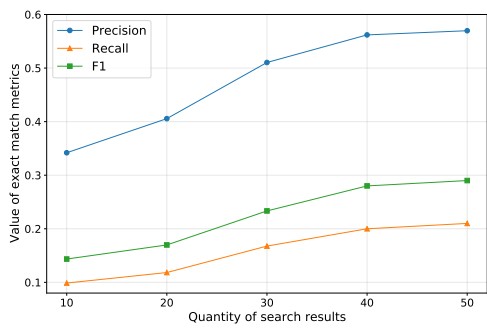

**Figure 4: Experiments with top results quantity.**

into two sets based on a length of 3. The experimental results are shown in Figure 3. The results indicate that most models have better performance on short queries (length ≤ 3). The reason is that parallel reformulation for longer queries requires more semantic constraints, which leads to greater generation difficulty. For example, "levi's **skirts**" could be an appropriate exploratory query of "levi's **shirts**". However, "levi's **skirts** for men" is absolutely an unfaithful and wrong exploratory query of "levi's **shirts** for men", because it ignores the gender constraint "for men". Besides, we observe that our proposed models (RPR, EQG, and LLMG) outperform baseline models significantly. This further confirms the effectiveness of using list structures for exploratory query generation.

### 5.4 Experiments with Number of Search Results

We use Top-50 search results in all experiments mentioned above. In this part, we further utilize different numbers of search results, ranging from 10 to 50, to investigate whether the quantity of search results can affect the performance of our rule-based model RPR. We use the metric Exact Match for comparison. Experimental results are shown in Figure 4. The figure shows that the quantity of search results does affect the quality of generated exploratory queries. As the quantity of search results increases, the quality of the generated exploratory queries improves. This is because more search results contain more list structures and thus generate more exploratory queries. Besides, more search results can also provide more evidence for ranking these exploratory queries and thus improve the quality of exploratory queries in the final output. The results also show that the improvement decreases when the quantity of search results increases, especially when it exceeds 40. This means that using top-50 results is enough.

### 5.5 Case Study

To compare the generated results of different models intuitively, we sample two real user queries and generate corresponding exploratory queries with different models. Table 3 shows the generated exploratory queries in groups, where BART (q) and BART (qs) are two baselines. The reformulated parts in queries are marked in bold. For the query "double java", BART (q) can hardly understand the meaning of the query and fails to generate other data types in java. With the support of snippets, BART(q+s) improves BART (q) but still generates irrelevant exploratory query "index java", which underperforms our proposed models RPR, EQG, and LLMG. Besides,

**Table 3: Examples of exploratory queries generated.**

| Query: **double** java | |
| --- | --- |
| Model | Exploratory queries |
| BART (q) | **3** java, **4** java, **2** java |
| BART (qs) | **index** java, **float** java, **string** java |
| RPR | **float** java, **long** java, **string** java |
| EQG | **float** java, **long** java, **int** java |
| LLMG (-list) | **quadruple** java, **multiple** java, **triple** java |
| LLMG | **float** java, **long** java, **int** java |

| Query: vests for **men** | |
| --- | --- |
| BART (q) | vests for **kids** |
| BART (qs) | vests for **kids** |
| RPR | vests for **kids**, vests for **brands** |
| EQG | vests for **kids**, vests for **kids & baby** |
| LLMG (-list) | vests for **kids**, vests for **women**, vests for **boys** |
| LLMG | vests for **kids**, vests for **women**, vests for **babies** |

without the list items as auxiliary information in the prompt, LLMG (-list) fails to understand the meaning of "double", generating some uncommon ("triple java") or even wrong results ("quadruple java"). For the second query "vests for men", its exploratory queries should be vests for other people. Both BART (q) and BART (qs) only generate "vests for kids", which lacks diversity. EQG improves RPR's "vests for brands" by generating "vests for kids & baby". Compared with other baselines, our LLM-based methods (LLMG (-list) and LLMG) generate more diverse exploratory queries such as "vests for women" and "vests for boys".

### 5.6 Limitations and Future Directions

Our work still has some limitations that we plan to address in future work. First, we divide the exploratory group simply based on the replaced terms in the query, which could be sub-optimal in some cases. For example, for an ambiguous query "go tutorial" and corresponding exploratory group "[java tutorial, c++ tutorial, chess tutorial, gomoku tutorial]", it would be better to divide the first two (programming language related) and last two (board game related) into different groups due to their different topics. Besides, in this work, we only focus on generating all possible exploratory groups without considering ranking them or generating corresponding exploratory questions. In fact, ranking these groups based on users' personalized interests or asking an exploratory question may further improve users' search experience.

### 6 CONCLUSION

In this paper, we propose generating exploratory queries to meet users' exploratory needs in conversational search. We first design a rule-based parallel reformulation model to generate exploratory queries based on list structures extracted from top retrieved documents. Then we propose to train a generative model in a multi-task learning manner for further generalization. Finally, we propose using LLMs for generation based on our well-designed prompts. We conduct several experiments on our annotated evaluation data and the experimental results not only validate the feasibility of utilizing list items to generate parallel queries but also demonstrate the effectiveness of the models we designed.

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
