# OpenReview forum: "Mining Exploratory Queries for Conversational Search"
_ACM.org/TheWebConf/2024/Conference — TheWebConf24 Oral_

### Official Review · Reviewer_uDYo · 2023-11-20

**Novelty:** 4
**Technical Quality:** 4

**Review:**

This paper centers on the generation of exploratory queries in the context of conversational search. It introduces a rule-based model designed to generate exploratory queries by leveraging information from the top-retrieved documents associated with the current query. Additionally, the paper puts forth a generative model that is trained using multi-task learning and incorporates a Language Model (LLM)-based generation approach with prompt engineering. To validate the effectiveness of these proposed methods, extensive experiments are conducted on a carefully constructed dataset.

Some concerns merit discussion
* Data Source Reliability:
The web documents acquired through the original query may not consistently cover a comprehensive range of parallel terms related to the query. Notably, terms such as "men" and "unisex," as well as "Rolex" and "Breitling," were absent in the top-retrieved documents when searching for "Cartier women watches".

* In EQG method, the rationale behind concatenating the masked query, masked term, and top-v list items as input to the BART encoder remains unclear, given their lack of an apparent sequential relationship.

* During experiments, the constructed dataset relies on input from three annotators, potentially introducing bias in the ground-truth exploratory queries for evaluation.

**Questions:**

* Q1:
The web documents acquired through the original query may not consistently cover a comprehensive range of parallel terms related to the query. Notably, terms such as "men" and "unisex," as well as "Rolex" and "Breitling," were absent in the top-retrieved documents when searching for "Cartier women watches".

* Q2:
 In EQG method, the rationale behind concatenating the masked query, masked term, and top-v list items as input to the BART encoder remains unclear, given their lack of an apparent sequential relationship.

* Q3:
During experiments, the constructed dataset relies on input from three annotators, potentially introducing bias in the ground-truth exploratory queries for evaluation.

**Reviewer Confidence:**

3: The reviewer is confident but not certain that the evaluation is correct

**Scope:**

3: The work is somewhat relevant to the Web and to the track, and is of narrow interest to a sub-community

---

### Official Review · Reviewer_8zby · 2023-11-22

**Novelty:** 6
**Technical Quality:** 5

**Review:**

The paper proposes new methods to produce Exploratory Queries, which are less studied with respect to clarifying queries or query expansion. The proposed model uses Large Language Models (LLMs) to produce the queries, and it obtains the better performances than those obtained by a rule-based system proposed in the model and to other BART-based models. The results of the proposed experiments are reported clearly in Table 2. For each measure the authors apply a t-test to verify if the improvements over different models are significant.

Pros of the paper:
1. Overall, the paper is well written and the proposed models are clearly described;
2. The addressed issue is relevant when surfing the web. It could help users during their search.
3. The proposed model achieves better results compared to the baselines. The results are significativly higher than the baselines.

Cons.: There are Minor Issues (M) and Limitations (L):
L1. There is a problem with the annotators since the reader cannot understand who they are, which is their educational level, if they have been paid for their work. The authors say that there are 3 annotators, are they enough for the evaluation?
L2. In Section 5.1, the authors describe the results without writing numbers: I suggest writing “our LLM-base model outperforms baseline of 0.2 in Precision, which is statistically significant …”, like in line 804. In this way, the effectiveness of the models is clearer. I suggest discussing the performances across different metrics and models. It is better to separate the Result table and the ablation one, because they refer to different paper sections. For the ablation study, the authors remove some features from RPR. I suggest
trying to add a unique feature to RPR incrementally, i.e. RPR with F^l, then RPR with F^l and F^c, ecc.

M1. At the end of the Abstract, I suggest briefly describing the dataset and the baselines, instead of writing
‘several baselines’ (lines 32 and 177).
M2. I suggest citing some more references in the paper. Especially, after the sentence that begins at line
109. I suggest citing the ‘previous studies of conversational search’. At lines 271-482, I suggest citing here
BART paper, which is cited in the following sections for the first time. I suggest citing some references for
Beam Search (line 555), Fleiss’ kappa (line 626) and BLUE measure (line 645). In section 3.3, in the
introduction, the authors write “Instruct an LLM”, which LLM? It should be written here. I suggest citing the
BART model using links to huggingface too.
M3. After the title of a section, I suggest writing what will be written in that section, e.g. in Section 2 the
authors should write “In this section we describe the related works divided into three main categories…”.
And, for me, it’s better to use “in this section/paragraph” instead of “in this part”, which is used a lot in the
paper (probably too much).
M4. The evaluation data READ ME does not explain the dataset, I suggest describing the dataset and its
structure in the READ ME too.
M5. In section 3.1.2, what are M,m and K? It could be understood by the reader, but it is better to specify. In
paragraph 3.1.3 (1), Instead of using A,B,C for defining function occ(), I suggest to be coherent with the
notation and using e,t,L_i. In Equations 2) and 4), do you mean the cardinality of the intersection set? If yes,
it should be |C(e) …|.
M6. Due to space limitations, I do not know if it is possible, but I suggest dividing the figure related to the
rule-based and LLM-based models in Figure 2.

**Questions:**

Have the authors tried RPR model with all the possible features couples? If yes, why their results are not
reported in the paper? If not, I suggest doing these experiments to complete the ablation study.

Who the evaluators are? What are their educational levels? Have they been paid for their work?

**Ethics Review Description:**

-

**Reviewer Confidence:**

3: The reviewer is confident but not certain that the evaluation is correct

**Scope:**

4: The work is relevant to the Web and to the track, and is of broad interest to the community

---

### Official Review · Reviewer_YJwh · 2023-11-24

**Novelty:** 4
**Technical Quality:** 4

**Review:**

This paper presents a well-conceived exploration of methods for generating exploratory queries in the context of conversational search engines. It offers a comprehensive view by comparing three approaches: heuristic rules (Rule-Based Parallel Reformulation, RPR), fine-tuned models (weakly supervised Exploratory Query Generation, EQG), and Large Language Model-based in-context learning (LLMG). This comparative perspective significantly enhances the work's value.

However, the paper could benefit from a clearer articulation of how these exploratory queries integrate with conversational search systems. This connection is crucial to understanding the broader application and impact of the research.

In terms of clarity, the paper leans towards an engineering focus, necessitating more explicit methodological details. Specific aspects, like the structure of the document collection used, require more comprehensive elaboration.

Regarding originality, this work stands out by introducing a novel retrieval-augmented approach to mining exploratory queries, a departure from the extensively studied areas of question suggestions and intent mining. This unique angle contributes to the paper's originality.

The significance of this work, while currently targeted at a niche application, holds potential for broader implications in conversational search. This broader impact could be more effectively realized with a stronger connection to conversational search applications.

Pros:
- The paper is well-structured and largely comprehensible.
- The experiments and ablation studies are well-executed, offering valuable insights into the effectiveness of the proposed methods.

Cons:
- Certain methodological details are insufficiently explained.
- The focus of the paper is somewhat narrow, concentrating on query generation rather than the empirical outcomes of searches.
- The evaluation would benefit from a more diverse range of exploratory query examples to better illustrate the effectiveness of the proposed methods.

In summary, while the paper is strong in its current form, enhancing the connection between exploratory queries and conversational search, along with expanding on method details and practical applications, could significantly elevate its impact and relevance.

**Questions:**

1. **Overall Impression**:
   - Has the paper validated its performance in conversational search, or does it only propose a method for exploratory query reformulation? Given the lack of empirical validation, should a more conservative tone be adopted when discussing its contributions to conversational search?

2. **Specific Concerns in Section 3**:
   - **[3.1.3 (1) & 3.1.3 (2) Query Reformulation Clarity]**: Can you clarify how the exploratory query candidates are generated? Specifically, does the number of potential queries equal the number of terms in the original query (\|q_0\|) multiplied by the number of terms in the parallel information (\|e\|)? Also, how does Equation (1) demonstrate the impact of reformulation on q0, and which terms in q0 are replaced? Is there a restriction on the number of terms replaced at a time?
   - **[3.1.3 (3) Mathematical Formulation of Candidate Queries]**: Could you mathematically define the set of candidate queries? What does "i" represent in Equation (3)? Also, does this function process all retrieved documents or the entire document corpus?
   - **[3.2.1 Text Generation Loss]**: In Equation (7), is there a standard text generation loss involved, and what is the purpose of the "min" operation in it? Are you optimizing for the minimum cross-entropy?
   - **[3.2.2 Leveraging Negative Terms]**: Was the use of negative (distracting) terms considered, and if so, were these terms minimized in likelihood? Would this approach align better with the objective of leveraging “conceptually inconsistent” terms?

**Reviewer Confidence:**

3: The reviewer is confident but not certain that the evaluation is correct

**Scope:**

4: The work is relevant to the Web and to the track, and is of broad interest to the community

---

### Official Review · Reviewer_csZb · 2023-11-27

**Novelty:** 2
**Technical Quality:** 2

**Review:**

This paper addresses the issue of search clarification and proposes a solution to meet users' exploratory needs. It introduces a rule-based model, a neural generation model, and a LLM to generate exploratory queries based on the current query's top retrieved documents. Experimental results demonstrate the effectiveness of these methods in generating high-quality exploratory queries comparing to the baselines.

Pros:
1.The paper addresses an important research problem.
2.Utilizing LLM through prompt learning is a current trend.
3.The paper is easy to follow.

Cons:
1.The novelty in this paper is limited.
2.The evaluation metrics proposed in Section 4.2 do not directly evaluate the quality of query recommendation, and they are also not used in the experiment result tables.
3.The ground truth data labeling lacks details.
4.The baselines do not include state-of-the-art approaches that generate exploratory queries.

**Questions:**

Outputting specific query extension options instead of exploratory options appears to be more of a system design choice rather than a technological challenge.

Can you include other methods used to expand exploratory queries as baselines? The current baselines seem to be weaker versions of the proposed method.

How did the authors obtain the ground truth data for the second task of their multi-task learning process?

Section 3.3 suggests that the process was conducted manually. Is this the case? How did they determine that the prompt could not be further improved?

The metrics defined in Table 1 seem ambiguous. Could you provide some metrics like the agreement ratio for the labeling part? Additionally, based on the labeling process, it seems that the recall metrics in Table 2 are not useful.

**Reviewer Confidence:**

4: The reviewer is certain that the evaluation is correct and very familiar with the relevant literature

**Scope:**

3: The work is somewhat relevant to the Web and to the track, and is of narrow interest to a sub-community

---

### Official Review · Reviewer_8jYf · 2023-12-02

**Novelty:** 5
**Technical Quality:** 4

**Review:**

The paper discusses the challenge of vague user queries and proposes a search clarification technique to address users' ambiguous search intents. While existing methods focus on refining the current query, users may also have exploratory needs diverging from their current intent. The paper introduces a method to mine exploratory queries as additional clickable options in conversational search systems. The approach involves a rule-based model to generate exploratory queries based on the top retrieved documents for the current query. The data generated by this model is then used to train a neural generation model through multi-task learning, further enhanced by leveraging a large language model for in-context learning. Extensive experiments demonstrate that the proposed methods generate higher-quality exploratory queries compared to several baselines, highlighting the usefulness of structure information in top retrieved documents for generating exploratory queries.

This work is timely and relevant to the WebConf series

the presentation is reasonable.

overall contribution is - a rule-based based approach 9rpr0 for mining exploratory queries, a BART encoder based approach (BART-EQG)and a zero-shot approach

The results include: RPR outperforms the two BART-based baselines and the EQG further improves RPR on all evaluation metrics
LLM-based method LLMG consistently outperforms all other methods and generate more diverse exploratory queries.

ablation studies conducted.

value of exact match perfromance increases as the number of search results increases.

For each query,..first ask a subject to manually write corresponding exploratory queries after a comprehensive survey on some resources (such as Wikipedia and top retrieved documents from Bing). - any guidelines used?

"to evaluate these exploratory queries in terms of three aspects: usefulness, faithfulness, and readability" - why these facets? any justification can be given?

**Questions:**

How generalisable are the rules? For example, will it work non-ecommerce scenarios?

"parallel reformulation for longer queries requires more semantic constraints" - Does this mean the effect of the methods is limited for real-life applications?

What are the computational complexity for any real-life exploitation?

"three annotators who understand our task well " - are they part of the authors or colleagues ?

For each query,..first ask a subject to manually write corresponding exploratory queries after a comprehensive survey on some resources (such as Wikipedia and top retrieved documents from Bing). - have you used any guidelines?

"to evaluate these exploratory queries in terms of three aspects: usefulness, faithfulness, and readability" - why these facets? any justification can be given?

How do you prove the reusability of the dataset?

Will you shared the dataset?

**Reviewer Confidence:**

3: The reviewer is confident but not certain that the evaluation is correct

**Scope:**

3: The work is somewhat relevant to the Web and to the track, and is of narrow interest to a sub-community

---

### Decision · Program_Chairs · 2024-01-22

**Decision:**

Accept (Oral)

**Comment:**

This paper focuses on search clarification, in particular mining exploratory queries to offer as suggestions in conversational search systems.

 The reviewers appreciated the importance of the research problem being tackled, but also had concerns about the novelty of the research and the experiments (results, ground truth, baselines). There were also some missing details on the methods and the practical value of the methods could also be better articulated. The concerns could generally be addressed with more explanation, clarification, and discussion, much of which the authors already provided in their rebuttal.

 In the rebuttal, the authors agree to address the concerns raised in the reviews, although without much detail in some cases (e.g., in the replies to reviewer YJwh (Part 1 of 2)) on how they would do that or what they would say. Without this knowledge, it is difficult to have full confidence that the revised version of the paper would satisfactorily address the concerns raised.

 In a separate comment, the authors also flagged a concern "regarding the review process" for feedback for one of the reviewers. Reading through the review from that reviewer and the authors' responses, I see this as no more than a disagreement on the value of the research between the authors and the reviewer (something that is healthy in peer reviewing) and not an issue with the process itself. In my opinion, the authors' responses to the reviewer's comments and questions do a reasonable job of addressing the concerns. The authors should make sure to integrate these responses in the next version of the paper as other readers may well have the same questions.